# Real-time MR-based 3D motion monitoring using raw k-space data

**Marius Krusen**                                               MA.KRUSEN@UNI-LUEBECK.DE

**Floris Ernst**                                               FLORIS.ERNST@UNI-LUEBECK.DE

*Institute for Robotics and Cognitive Systems, University of Lübeck, Lübeck, Germany*

**Editors:** Accepted for publication at MIDL 2024

## Abstract

Due to its great soft-tissue contrast and non-invasive nature, magnetic resonance imaging (MRI) is uniquely qualified for motion monitoring during radiotherapy. However, real-time capabilities are limited by its long acquisition times, particularly in 3D, and require highly undersampling k-space resulting in lower image resolution and image artifacts. In this paper, we propose a simple recurrent neural network (RNN) architecture to continually estimate target motion from single k-space spokes. By directly using the incoming k-space data, additional image reconstruction steps are avoided and less data is required between estimations achieving a latency of only a few milliseconds. The 4D XCAT phantom was used to generate realistic data of the abdomen affected by respiratory and cardiac motion and a simulated lesion inserted into the liver acted as the target. We show that using a Kooshball trajectory to sample 3D k-space gives superior results compared to a stack-of-stars (SoS) trajectory. The RNN quickly learns the motion pattern and can give new motion estimations at a frequency of more than 230 Hz, demonstrating the feasibility of drastically improving latency of MR-based motion monitoring systems.

**Keywords:** MR-guided radiotherapy, motion estimation, real-time MRI, recurrent neural network, XCAT phantom, MR-linac, undersampling

## 1. Introduction

In recent years, the development of MR-linacs (Lagendijk et al., 2014; Mutic and Dempsey, 2014) has opened up new avenues for great advancements in the field of radiation oncology. An MR-linac combines a linear accelerator with magnetic resonance imaging (MRI) to allow online adaptation of the treatment plan before and even during the treatment with a radiation beam, giving rise to MR-guided radiotherapy. Compared to computed tomography (CT), the predominant imaging modality in radiotherapy, MRI offers superior soft-tissue contrast and does not expose the patient to ionizing radiation (Pereira et al., 2014). It also does not require a highly invasive fiducial marker implantation unlike many X-ray imaging-based methods (Tang et al., 2007; Fledelius et al., 2014; Bertholet et al., 2017).

One of the critical challenges in realizing the full potential of MR-guided radiotherapy is the fast and accurate monitoring of intrafraction motion. Due to respiration, cardiac motion and other factors, the position of the target volume can change significantly during the treatment. For the liver, motion amplitudes of up to 30 mm have been reported (Park et al., 2012). This can lead to underdosage of the target volume and overexposure of surrounding healthy tissue. Tracking the target in real time can mitigate this problem, reducing the risk of complications and improving treatment outcome (Al-Ward et al., 2018).

However, acquiring high resolution 3D MRI data typically takes several minutes, which is too slow for real-time motion monitoring.

Since many MR-based motion monitoring methods rely on MR images, they have to work on lower resolution or fewer images, like using interleaved 2D slices (Bjerre et al., 2013; Tryggestad et al., 2013), or reconstruct images from highly undersampled k-space data (Mickevicius and Paulson, 2017) to achieve a high enough speed. This can be combined with other MRI acceleration techniques like parallel imaging or compressed sensing to further reduce acquisition times (Tsao and Kozerke, 2012). Deep learning-based reconstruction methods have also been proposed to further improve the speed and quality of the reconstructed images (Hammernik et al., 2017; Muckley et al., 2021). Few approaches work directly on k-space data, including k-space-based navigators (Kober et al., 2012) and methods using the phase correlation method to register measured data to some reference k-space data (Vaillant et al., 2014). This is often combined with a patient-specific motion model built during a less time-critical training phase before the treatment (Liu et al., 2021; Keijnemans et al., 2022). Huttinga et al. (2020) were able to reduce the latency of their MR-MOTUS framework from around 5 minutes for 3D motion to 170 ms by also splitting the reconstruction problem into an offline preparation and an online inference phase (Huttinga et al., 2022).

Our approach is to sample the k-space with a radial trajectory and directly use the incoming radial k-space lines (spokes) to continually estimate the target motion using a recurrent neural network (RNN). This way, no additional image reconstruction steps are performed and new motion estimations can be given after the acquisition of each new spoke in just milliseconds. In a previous work, we demonstrated this approach on a 2D rigid head motion estimation task achieving sub-millimeter accuracy (Krusen and Ernst, 2024). Motion was simulated by rigidly transforming slices of 25 MRI head scans and the k-space was sampled using a golden-angle radial trajectory. Additionally, it was shown that removing the peripheral values of each spoke increases generalization across different patients.

Here, we extend this approach to a more realistic 3D motion estimation scenario. The 4D XCAT phantom (Segars et al., 2010) is used to generate realistic abdominal MRI data affected by respiratory and cardiac motion. Since XCAT is a numerical phantom, there are no interpolation inaccuracies in the generated volumes and k-space sampling can also be done without interpolation. This sampling of the 3D k-space is done using a kooshball trajectory (Chan et al., 2009), the 3D generalization of the golden-angle trajectory, and compared to a stack-of-stars (SoS) trajectory (Chandarana et al., 2011), which uses golden-angle sampling along the $k_x$-$k_y$ plane but Cartesian sampling along the $k_z$ dimension. A slightly adapted RNN is trained on the generated data to track the motion of a spherical tumor inserted into the liver.

## 2. Material and Methods

### 2.1. Motion and MRI simulation

The motion and image data of the abdominal region was mainly generated using the XCAT phantom (Segars et al., 2010). The 4D XCAT phantom is a numerical phantom supporting a number of anatomic models of male and female bodies with thousands of anatomical

structures that simulates cardiac and respiratory motion at any spatial resolution and offers both deformable vector fields and image volumes as output. For these MRI motion datasets the resolution was set to $1\,\text{mm} \times 1\,\text{mm} \times 1\,\text{mm}$ resulting in volumes of size $320 \times 320 \times 320$ centered around the liver. Ten different anatomic models were used to create a dataset and for each model ten sequences of $1\,000$ time steps were generated. The length of one time step was set to $4\,\text{ms}$ to simulate a steady-state spoiled gradient echo sequence with a repetition time (TR) of $4\,\text{ms}$.

For each sequence a spherical tumor with a random diameter between 10 and $20\,\text{mm}$ was inserted at a different location inside the liver. To further increase variance between sequences, different heart and respiration rates were used, and heart and respiration rate variability were simulated to vary the length of cardiac and respiration cycles during a sequence as well. An average resting heart rate of around 65 beats per minute and an average free-breathing respiration rate of around 13 breaths per minute, as well as the variability values were taken from Schumann et al. (2021) and Kobayashi (2021) respectively. Additionally, the initial phase of the cardiac and respiratory cycle at the beginning of a sequence was randomized and the two XCAT parameters controlling the maximum diaphragm motion and the maximum expansion of the chest along the anteroposterior axis were varied after every breath.

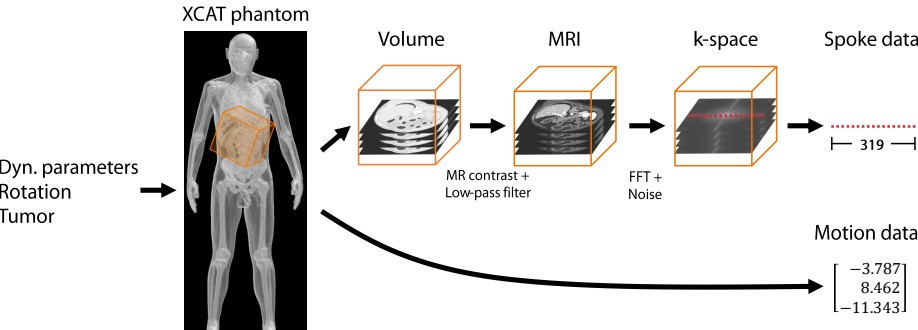

Figure 1: The XCAT data generation pipeline. Dynamic parameters, like the heart and respiration rates, the desired rotation and tumor information are given to the XCAT phantom to generate image volumes and motion data for each time step. MR contrast and a low-pass filter are applied to the volumes before transforming them to k-space and adding noise. Finally, the k-space is sampled resulting in the spoke data for the dataset.

Simulating k-space spoke data from the image volumes generated with the XCAT phantom involved multiple post-processing steps as shown in Figure 1. First, MR contrast is applied to the volumes. The pixel intensities of different organs and tissues for this step were extracted from T1-weighted MR images from the Duke Liver Dataset (Macdonald et al., 2020) and randomly varied between sequences. Additionally, a low-pass filter was applied to blur the pixelated edges between tissues. The resulting MR image volumes were transformed to k-space using a fast Fourier transform and complex-valued white Gaussian

noise was added to achieve a signal-to-noise ratio of 20. Finally, the k-space was sampled following a kooshball trajectory for one dataset and following a stack-of-stars trajectory for a second dataset. Both are radial trajectories that would require interpolation. Since the XCAT phantom allows generating the image volumes at arbitrary rotations, each volume was already rotated during generation according to the desired k-space trajectory to make sure that the sampled spokes will align with the grid of the image volumes and no interpolation is necessary. The result is a sequence of 1 000 k-space spokes of 319 sampling points each.

The tumor motion simulated by the XCAT phantom by default is a translation along the sagittal axis (y-axis) and longitudinal axis (z-axis) based on the respiratory phase and parameters like the maximum diaphragm motion and the maximum expansion of the chest. To have motion in all three dimensions, a custom sinusoidal motion trajectory was used that resembles the default motion along the y-axis and z-axis but also includes a smaller motion along the x-axis. On average, the motion amplitude along the x-axis is 30% and along the y-axis is 50% of the amplitude along the z-axis. Cardiac motion is simulated and affects the overall image volumes and k-space data, but does not contribute to the motion of the tumor. The resulting motion data is a sequence of translational motion along all three axes.

In addition to the two datasets for both trajectories with a TR of 4 ms, another one with a TR of 40 ms is generated for each, for a total of four datasets. The higher TR allows longer simulated sequences spanning multiple cardiac and respiratory cycles without generating much longer sequences in this computationally expensive process.

### 2.2. Network training

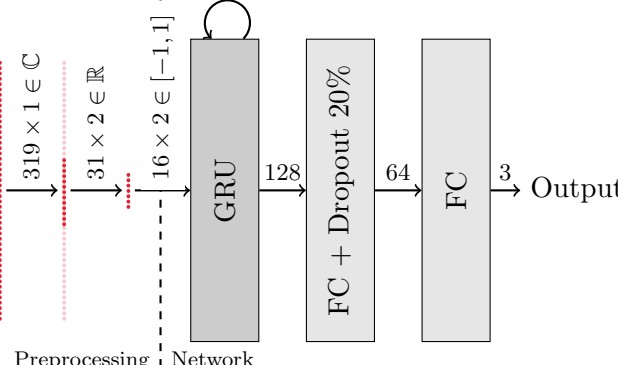

Figure 2: The preprocessing steps and network architecture. A complex-valued input spoke is split into real and imaginary data, shortened to the central 10 %, scaled down and its two halves are averaged during preprocessing. The gated recurrent unit (GRU) layer learns the temporal relation of the data before the fully connected (FC) layers with ReLU activation and dropout in between give the motion estimation.

Before passing through the network, the input data goes through a few preprocessing steps. First, the complex-valued spoke data is split into its real and imaginary parts, since most machine learning methods only work on real-valued input data. Then, peripheral data of the spoke is removed leaving only the central 10 %. This improves generalization and leads to a more stable and fast training, as previously shown (Krusen and Ernst, 2024). To further improve generalization, the average of the two symmetric halves of the spoke is used. Finally, the data is scaled by a constant factor that is sufficiently large, so that all current and future values are most likely in the range of $[-1, 1]$.

During training two methods are used to increase the amount and variety of training data. The first method is subsequence learning. For each sequence of 1 000 time steps in the dataset there are many subsequences that can also be used for training. In particular, all 999 subsequences that only start later than the full sequence provide distinct and new data, since the start condition changes, so both the progression of the internal state of the GRU is different and the target motion is relative to a different initial position. Out of those, the 500 subsequences that are at least 500 time steps long are used as additional training data.

The second method is applying different augmentations to the training data in each epoch. This gives more variety to the data during training, especially since the subsequence learning introduces many repeat inputs of the same data which can quickly cause overfitting. Three different augmentations are applied to the input data every time before passing it into the network. First, each spoke is scaled by a random factor between 0.9 and 1.1. Since this only alters the magnitude, the resulting spoke is still a valid input for the resulting motion target. Second, each individual value is multiplied by a factor sampled from a normal distribution with a mean of 1 and a standard deviation of 0.1. This can be seen as additional noise that changes every iteration. Lastly, there is a 5 % chance for a spoke to be swapped with the next one in a sequence. While this slightly invalidates the target values, the time between subsequent spokes is short enough that the changes to the target values would be miniscule. Furthermore, a noise augmentation is also applied to the target data by adding a value sampled from a normal distribution with a mean of 0 and a standard deviation of 0.1.

Network training is done using the Adam optimizer (Kingma and Ba, 2014) and a learning rate of 0.001 that decays with a factor of 0.75 every 5 epochs for 25 epochs. All subsequences of same length are processed in one batch resulting in a batch size of 80. The mean square error between estimated and actual motion at each time step is used as the loss function. The training was performed on an NVIDIA A100 using the Pytorch framework and took about one hour.

## 2.3. Evaluation

The trained networks are evaluated using a 5-fold cross validation. Each fold consists of the 10 full sequences of two different body models, resulting in 20 sequences per fold. The mean absolute error (MAE) between the estimated and actual motion at each time step is used as the evaluation metric. The average MAE of all sequences and folds is then used to compare the performance of the networks.

Furthermore, the *jitter* of the output signal is calculated to determine how smooth the output is. This is an important quality for actually compensating the estimated motion,

since a jittery output could exceed the mechanical capabilities of a subsequent multileaf collimator (MLC) for example. The jitter is calculated as the first derivative of the error between the estimated and actual motion. The mean absolute jitter then quantifies how much the output signal oscillates around the target. To reduce the jitter and smoothen the output, a simple moving average filter can be applied to the output signal. A filter length of 3 implies delaying the output signal by 1 time step and is used here to evaluate the effect on the jitter and the MAE.

The network trained on the Kooshball dataset is compared to the network trained on the stack-of-stars dataset for both a TR of 4 ms and 40 ms. To put their scores into context, two simple additional estimators are used. The first one is a constant estimator that always outputs the mean motion of the target over the whole sequence to show the best possible error if no motion estimation is performed. The second one is a hypothetical perfect estimator with latency, whose output is the exact target function but delayed by 200 ms. 200 ms is the maximum suggested latency for real-time motion monitoring (Murphy et al., 2002) and comparable to existing approaches like MR-MOTUS (Huttinga et al., 2022). This estimator shows the impact of just latency on the error. The reported error and uncertainty that real approaches have is not considered here and would further increase the total error.

## 3. Results

The inference time of the trained networks is around 0.3 ms on an NVIDIA GeForce GTX 970, making the TR the main factor for the latency of the approach. Assuming a TR of 4 ms, the network can give new motion estimations at a frequency of more than 230 Hz.

Table 1: Quantitative results of the 5-fold cross validation. The mean absolute error (MAE) and the jitter of the output signal are shown for two different trajectories and two different TRs. The results of two simple additional estimators are shown for comparison, a constant estimator and a perfect estimator with a latency of 200 ms.

| Method | TR (ms) | x MAE (mm) | y MAE (mm) | z MAE (mm) | Jitter |
|---|---|---|---|---|---|
| RNN (Kooshball) | 4 | 0.56 ± 0.48 | 0.95 ± 0.80 | 1.86 ± 1.57 | 0.076 |
| RNN (SoS) | 4 | 0.96 ± 0.83 | 1.59 ± 1.34 | 3.18 ± 2.69 | 0.071 |
| Const. mean | 4 | 1.89 ± 1.11 | 3.14 ± 1.86 | 6.31 ± 3.72 | 0.024 |
| Perfect w. delay | 4 | 0.59 ± 0.45 | 0.98 ± 0.77 | 1.96 ± 1.51 | 0.009 |
| RNN (Kooshball) | 40 | 0.64 ± 0.57 | 1.07 ± 0.93 | 2.13 ± 1.88 | 0.185 |
| RNN (SoS) | 40 | 2.11 ± 1.88 | 3.47 ± 3.07 | 6.89 ± 6.07 | 0.330 |
| Const. mean | 40 | 1.88 ± 1.05 | 3.15 ± 1.76 | 6.32 ± 3.51 | 0.231 |
| Perfect w. delay | 40 | 0.57 ± 0.41 | 0.96 ± 0.69 | 1.92 ± 1.38 | 0.081 |

The results of the cross validation are shown in Table 1. The RNN trained on the Kooshball dataset has a much lower MAE than the RNN trained on the stack-of-stars dataset for both TRs. For the more realistic TR of 4 ms, the Kooshball RNN has an MAE of 0.56 mm, 0.80 mm and 1.86 mm along the x-axis, y-axis and z-axis, respectively, slightly

outperforming the perfect estimator with a delay. Appendix A shows how the MAE varies greatly between sequences and during a respiration cycle, explaining the large confidence intervals. The dataset with a TR of 40 ms causes both RNN methods to perform worse, while the two additional estimators expectedly perform similarly. In particular, the SoS RNN got even worse than the constant estimator anymore, suggesting that it does not learn much anymore. Appendix B shows that the subsequence learning and using half spokes during preprocessing improve the accuracy of the RNN. Applying a simple moving average filter to the output of the Kooshball RNN reduces the jitter to 0.030 while only slightly increasing the MAE to 0.57 mm, 0.96 mm and 1.87 mm, respectively. The SoS RNN shows a similar result. Smoothening the output for the higher TR of 40 ms would also reduce the jitter, but due to the longer delay, the MAE would also see an undesirable increase.

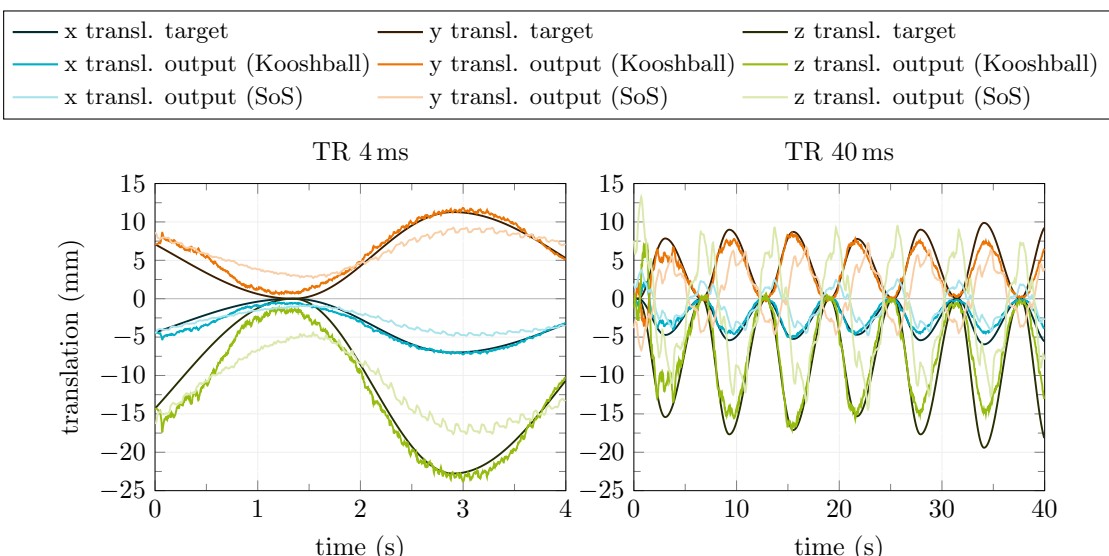

Figure 3: Example motion sequences. The output of the Kooshball RNN follows the target well but underestimates the peaks, mainly for a TR of 40. The SoS RNN produces a less accurate output with an oscillation caused by the start of a new stack.

The sequences in Figure 3 show example outputs of both RNN methods for both TRs overlaid on the target motion. The Kooshball RNN is able to follow the target motion accurately but often underestimates the peaks of the target motion. For the SoS RNN this behaviour can also be seen, albeit less accurately and with a noticeable oscillation with a period of around 20 time steps. This is exactly the number of slices in a stack in the dataset, showing one of the problems of the SoS trajectory for continuous motion estimation.

## 4. Discussion

This work presents an learning-based approach that can rapidly estimate 3D motion from raw k-space data. The 4D XCAT phantom was used to generate abdominal MRI data

affected by realistic respiratory and cardiac motion. A spherical tumor is inserted into the liver to provide a tracking target for the motion estimation. As a first step and due to the limits of the XCAT phantom, the tumor does not rotate or deform during motion unlike some of the other generated organs. However, these effects are typically small compared to the overall displacement of a tumor (Xu et al., 2014) and the phantom offers a reproducible way to generate volume data together with the corresponding motion data at a temporal resolution that is not feasible with real data.

The presented RNN learns the motion pattern of the tumor to quickly give accurate motion estimations. While the accuracy is not as high as in the rigid 2D case (Krusen and Ernst, 2024), the MAE is still small and allows for smaller planning target volumes and shorter treatment times. A hypothetical perfect estimator with a latency of 200 ms already has a similar error just due to latency and it does not yet take into account the actual tracking error that existing comparable approaches have. Improvements to the accuracy could be achieved by simulating more varied motion, possibly in combination with a more complex network architecture. Due to the periodic nature of respiratory motion the RNN could learn the general respiration pattern and not taking variations between respirations into account, as example outputs in Figure 3 could suggest. Training on more varied and less periodic data worked well for the 2D case (Krusen and Ernst, 2024) and could not just improve accuracy but also allow accurate motion estimation of other non-periodic intrafraction motion types, like slow drifting motion or patient motion.

The Kooshball trajectory was shown to be superior to the stack-of-stars trajectory for this task. As the 3D generalization of the golden-angle trajectory, it provides a faster and more even coverage of 3D k-space avoiding artifacts like the oscillation seen in the SoS output and resulting in a much lower MAE.

The main advantage of the approach is its very low latency. The effect that even a delay of just 200 ms has, which is considered enough for real-time motion monitoring, can be seen in the result of the hypothetical perfect estimator. With a estimation frequency of more than 230 Hz, the approach can apply a simple moving average filter to smoothen the output without an impact on the MAE. Additionally, predictions into the future, which are necessary in a motion compensation system to account for the system latency, are much more accurate with a short latency.

## 5. Conclusion

In this work, we present an RNN approach to continuously estimate 3D motion from raw k-space data. By directly using the k-space data coming in from a 3D radial trajectory, the network can estimate motion with minimal latency at a frequency of more than 230 Hz. To train and validate the approach, the 4D XCAT phantom was used to generate realistic abdominal MRI data with an inserted tumor affected by respiratory and cardiac motion. The Kooshball trajectory was shown to be superior to the SoS trajectory for this task, since it more evenly covers the entire 3D k-space. The speed of the approach allows smoothening the output with a simple moving average filter and demonstrates the feasibility of a very low-latency MR-based motion monitoring system. Future work will focus on further improving the accuracy of the approach.

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

## Appendix A. Error analysis

Figure 4 shows the errors, averaged over all three axes, for the 10 sequences of one body model with a TR of 4 as box plots. It can be seen that our RNN-based method and the hypothetical perfect estimator with delay perform similar on most sequences, like their

similar overall MAE suggests. However, the variation in MAE between sequences is quite high, explaining the large confidence intervals of all methods in Table 1. The large number of outliers of the RNN-based method seen in sequences 2 and 7 stems from uncertainty at the beginning of the sequences. In the initial time steps, when the RNN did not receive much data yet, the output can sometimes be less or even more accurate than during the rest of the sequence.

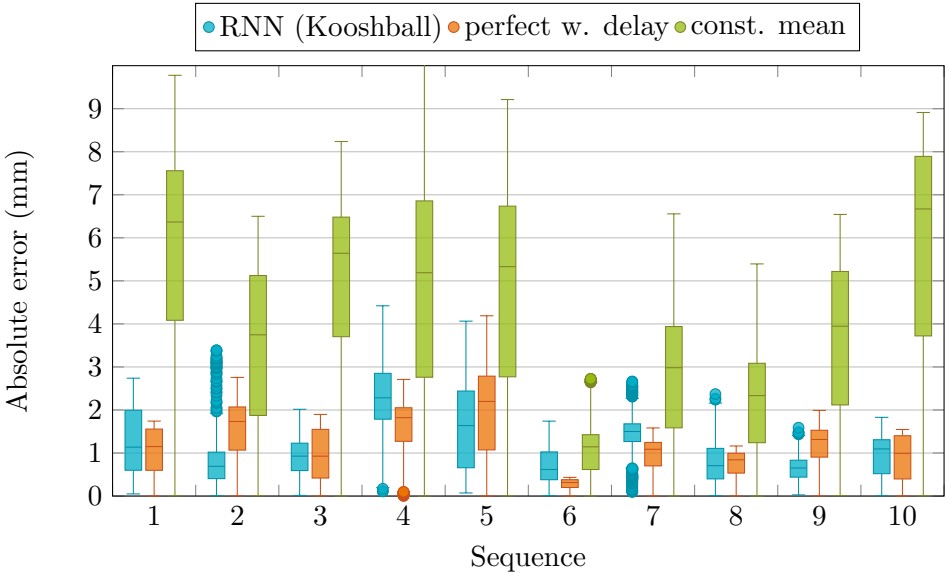

Figure 4: Box plots showing the absolute errors of the 10 sequences with a TR of 4 of a single body model for three methods.

Another cause of variation in error is the respiration phase. As seen in Figure 5, the mean absolute error varies during a respiration cycle. The value for the current respiration phase used by the XCAT phantom is scaled between 0 and 1, with 0 being full exhale and 0.4 being full inhale. The RNN has the largest errors around the full inhale phase when the motion amplitudes is largest which is typical for motion estimation methods. The delayed perfect estimator on the other hand has much larger variations during a respiration cycle and the smallest errors around the full inhale phase. This is due to the speed of motion being the slowest around the full inhale and full exhale phases, whereas the speed is much faster and the errors much larger inbetween these phases. Since the perfect estimator is delayed it is more sensible to fast motion.

## Appendix B. Ablation studies

To investigate the effect of certain components of the approach on the accuracy, an ablation study is performed. The results can be seen in Table 2. In a first experiment, the full k-space spokes are used as input to the RNN instead of the average of the two symmetric halves of

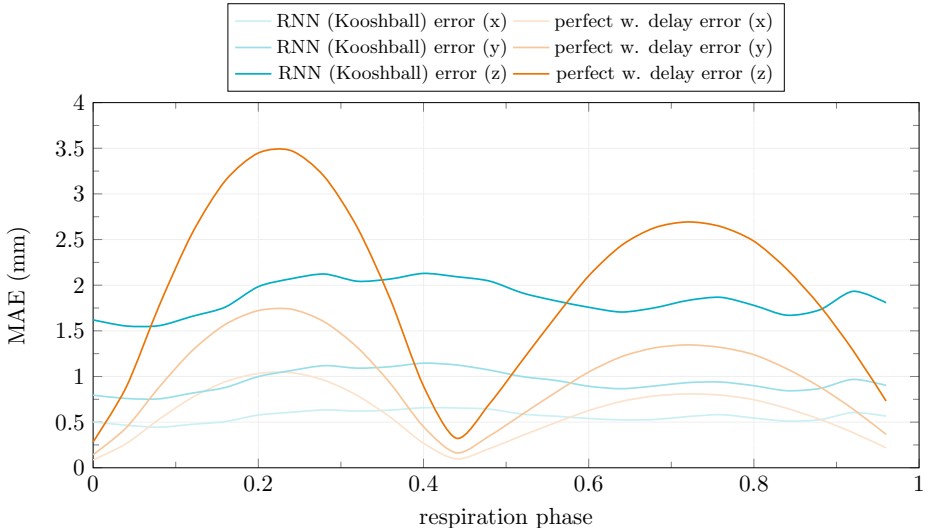

Figure 5: The mean absolute error over the respiration phase. The phase is scaled from 0 to 1, with 0 being full exhale and 0.4 being full inhale.

every spoke. The resulting MAE is slightly higher than the presented RNN, especially for a TR of 4, confirming the advantage of this preprocessing step.

The second experiment evaluates the effect of subsequence learning by training the RNN without using subsequences. To keep results somewhat comparable, the training without subsequences runs for 1000 epochs instead of 25 and the learning rate decays every 200 epochs instead of every 5. While this still results in less total sequences processed during training, the results are already pretty clear with no real improvement happening during most of the training. The MAE for these experiments is much higher than that of the presented RNN. The dataset with a TR of 4, which also has much less variation in the target motion, is particularly affected by this and leads to overfitted results worse than the constant estimator.

Table 2: Quantitative results of the ablation study. The mean absolute error (MAE) and the jitter of the presented RNN are compared to those of an RNN trained with full k-space spokes as input and one trained without subsequences for two different TRs.

| Method | TR (ms) | x MAE (mm) | y MAE (mm) | z MAE (mm) | Jitter |
|---|---|---|---|---|---|
| RNN (Kooshball) | 4 | $0.56 \pm 0.48$ | $0.95 \pm 0.80$ | $1.86 \pm 1.57$ | 0.076 |
| full spokes | 4 | $0.66 \pm 0.54$ | $1.13 \pm 0.92$ | $2.19 \pm 1.81$ | 0.076 |
| no subsequences | 4 | $1.54 \pm 1.40$ | $2.63 \pm 2.40$ | $5.29 \pm 4.79$ | 0.190 |
| RNN (Kooshball) | 40 | $0.64 \pm 0.57$ | $1.07 \pm 0.93$ | $2.13 \pm 1.88$ | 0.185 |
| full spokes | 40 | $0.67 \pm 0.62$ | $1.12 \pm 1.03$ | $2.21 \pm 2.04$ | 0.174 |
| no subsequences | 40 | $1.42 \pm 1.15$ | $2.38 \pm 1.94$ | $4.78 \pm 3.84$ | 0.452 |

