# OpenReview forum: "Real-time MR-based 3D motion monitoring using raw k-space data"
_MIDL.io/2024/Conference — MIDL 2024 Poster_

### Official Review · Reviewer_ZGD6 · 2024-02-27

**Confidence:** 3
**Preliminary Rating:** 3
**Final Rating:** 5

**Summary:**

This work proposes to use a recurrent neural network (RNN) to predict motion during MRI acquisition directly from k-space. The sequence of radial k-space samples are used as timepoints, and a motion prediction (in terms of difference with t=0) is output at every step. Predicting directly from undersampled k-space saves time allowing theoretical real-time motion correction during radiotherapy, for example. The paper builds upon previous work from the authors by exploring new sampling schemes on 3D data, instead of 2D.

**Strengths:**

The paper is extremely very well written. Everything is well presented, easy to understand and well motivated. The method is simple yet effective, the experimental protocols are well defined, the datasets considered are interesting and relevant.

**Weaknesses:**

The paper feels very incremental with little added value over previous work. Even though an additional dimension is added to the dataset, the network still predicts 2D motion. The results are somewhat disappointing, unfortunately.

**Detailed Comments:**

The paper is well written and should be taken as an example of good writing. The text is perfectly structured, concise yet nothing is omitted. The figures are relevant, well constructed. The paper follows a nice structure and the content of each section is on topic. The method proposed is simple and interesting. While the work is incremental, it does not require prior knowledge of the previous work. The method is well motivated, the problem as posed is interesting and relevant.

Unfortunately, as mentioned, the proposed method feels a little too incremental. While the main contribution is the dataset going from 2D to 3D, the network architecture and task at hand is not really different as this work predicts y-z motion while prior work predicted x-y motion.

While lack of novelty should not be a reason for refusal in itself, the results as presented are also a bit disappointing. Two “degenerate” methods are presented as lower and upper performance bounds, and the (best version of the) proposed work kind of sits in the middle. It is possible however that I have misunderstood the results. In this case they would benefit from being presented in a more positive manner.

**Justification Of Final Rating:**

Considering initial comments and the subsequent discussion, the authors have done a tremendous job of addressing all concerns raised (for example on the similarity with prior work or the results obtained from this work) by extending their proposed work and reworking the text to clear up any confusion.

**Justification Of The Preliminary Rating:**

While the paper is exceptionally well written, it may also just not be exciting enough to warrant a publication at MIDL. As mentioned previously, while a lack in novelty should not be grounds for rejection, a lack in both results and novelty makes it hard to consider.

**Questions To Address In The Rebuttal:**

Please put forward the distinctions with prior work and explain the seemingly mild results.

**Special Issue:**

No

---

> ### Author Response · Authors · 2024-03-17
>
> Dear reviewer,
>
> thank you very much for your helpful feedback and your comments on the writing style!
>
> We agree that the paper is incremental, especially the network architecture.
> However, we believe that the step from 2D to 3D is a significant change.
> Additionally, the target motion changed from being an entirely rigid transformation of the whole field of view to the motion of a tumor surrounded by deforming organs with potentially different motion patterns.
> The default tumor motion generated by the XCAT phantom is limited to two axes.
> However, we have now modified the tumor motion to also include periodic motion in the x-axis based on the respiratory phase.
> Since the data generation is a time-consuming process, we have so far only been able to generate datasets with this motion pattern for the Kooshball trajectory but not the stack-of-stars trajectory.
> The preliminary results of training on these datasets are not just comparable, but actually seem to improve the accuracy of the approach, at least in the y-direction.
> The data generation will not be finished in time for the rebuttal, but we would like to include these results in the camera-ready version of the paper if they do not change the general sentiment of the paper.
>
> In regard to the "somewhat disappointing" results, we have clarified the perfect estimator comparison to put our results better into context.
> Considering the additional error that existing methods have just due to delay highlights the speed advantage of our method and should present the results in a more positive manner.
> If we assume that the total error from latency and tracking is not entirely additive, we can conservatively estimate that the total error of algorithms like MR-MOTUS will likely be larger than 2 mm.
> Thank you again for your comments and taking the time to review our work.

---

> > ### Comment · Reviewer_ZGD6 · 2024-03-18
> >
> > Thank you for your reply.
> >
> > > The default tumor motion generated by the XCAT phantom is limited to two axes. However, we have now modified the tumor motion to also include periodic motion in the x-axis based on the respiratory phase. Since the data generation is a time-consuming process, we have so far only been able to generate datasets with this motion pattern for the Kooshball trajectory but not the stack-of-stars trajectory. The preliminary results of training on these datasets are not just comparable, but actually seem to improve the accuracy of the approach, at least in the y-direction. The data generation will not be finished in time for the rebuttal, but we would like to include these results in the camera-ready version of the paper if they do not change the general sentiment of the paper.
> >
> > While the inclusion of this dataset would have been a great addition to the paper, cementing the 3D aspect of the proposed method (as I presume x motion would now also have been predicted), I unfortunately cannot base my review on future results.
> >
> > > In regard to the "somewhat disappointing" results, we have clarified the perfect estimator comparison to put our results better into context. Considering the additional error that existing methods have just due to delay highlights the speed advantage of our method and should present the results in a more positive manner.
> >
> > Unfortunately, as far as my comprehension of the paper goes, the takeaway from the results as presented remain similar. The proposed method using Kooshball trajectories remains quite worse than the perfect estimator with maximum delay, with much higher variance (as highlighted in Figure 4, looking at the number of outliers). The speed of the proposed method seems to be its main appeal (c.f section 4: "The main advantage of the approach is its very low latency."), however section 3 mentions "making the TR the limiting factor for the latency of the approach". The proposed method therefore seems to be much faster than it needs to be (0.3ms vs 4ms), and would maybe benefit from a bigger model, more expressive model ?

---

> > > ### Author Response · Authors · 2024-03-22
> > >
> > > Thank you for your quick response.
> > > We understand that unfinished results can and should not impact the review.
> > > However, the data generation and training has since been completed, and we have revised the paper to include the results of the new datasets.
> > > We could confirm our preliminary results and the new datasets not only solidify the 3D aspect of our method but also improve its accuracy.
> > > We have elaborated on this a bit more in our general response to all reviewers.
> > >
> > > The method being much faster than it needs to be is only partly true.
> > > The *frequency* of around 230 Hz of the motion estimation would be unaffected by a bigger model, that would make use of the those 4 ms.
> > > However, the *latency* between motion occurring and being estimated by our method would increase.
> > > We have changed the text to call the TR "the *main* factor for the latency" instead.
> > > Still, this very low latency leaves room for a more complex and expressive model without latency-induced errors becoming a larger problem and is something we want to explore in the future.

---

> > > > ### Comment · Reviewer_ZGD6 · 2024-03-25
> > > >
> > > > > Thank you for your quick response. We understand that unfinished results can and should not impact the review. However, the data generation and training has since been completed, and we have revised the paper to include the results of the new datasets. We could confirm our preliminary results and the new datasets not only solidify the 3D aspect of our method but also improve its accuracy. We have elaborated on this a bit more in our general response to all reviewers.
> > > >
> > > > Thank you for addressing this concern. The revised method and results, in my opinion, shine a brighter light onto the proposed work and truly distance it from prior work.
> > > >
> > > > > The method being much faster than it needs to be is only partly true. The frequency of around 230 Hz of the motion estimation would be unaffected by a bigger model, that would make use of the those 4 ms. However, the latency between motion occurring and being estimated by our method would increase. We have changed the text to call the TR "the main factor for the latency" instead. Still, this very low latency leaves room for a more complex and expressive model without latency-induced errors becoming a larger problem and is something we want to explore in the future.
> > > >
> > > > Thank you for correcting my misunderstanding of this aspect of the proposed method and adapting the text to prevent other readers from making the same mistake.
> > > >
> > > > Considering the two discussion points above, I have no further comments and will be revising my initial rating.

---

### Official Review · Reviewer_app7 · 2024-02-28

**Confidence:** 3
**Preliminary Rating:** 4
**Recommendation:** Poster
**Final Rating:** 4

**Summary:**

This paper extends a previous works from 2d rigid head motions to realistic 3d trajectories of abdominal magnetic resonance imaging subject to cardiac and respiratory motion. The author leverage RNN to design an algorithm able to track a tumor in a liver from data recorded with a phantom, and show reasonable accuracy for potential use in the clinic.

**Strengths:**

The paper is well-written, well-structured and with clear motivation, experiment and algorithm descriptions. The algorithm seems sound and produces decent results, with a margin of refinement and potential use cases on patient-based datasets.

**Weaknesses:**

I did not see any major weaknesses in this work. Maybe one could expand a bit on how the present work extend the previous work of the author, which is only mentioned in the introduction rapidly. Also the application of this method is only on a single, home made dataset, and it could be interesting to apply it to other, known dataset with reference, or benchmark results. I am not versed enough in this topic to propose/point to some, but if this is possible, it could be a nice addition to showcase the accuracy of this work. I would see this as a bonus only.

**Detailed Comments:**

I don't have ay specific comments.

**Justification Of Final Rating:**

Very good, I do not have any further comments.                                                                                                                                                            .

**Justification Of The Preliminary Rating:**

As this paper is "only" an extension of a previous work, and does not extensively compare the accuracy with benchmark datasets, or apply it to patients with more noise, etc... there is still some work to make this algorithm fully applicable, but its promising!

**Questions To Address In The Rebuttal:**

I have no questions.

**Special Issue:**

No

---

> ### Author Response · Authors · 2024-03-17
>
> Dear reviewer,
>
> thank you for your comments and ideas on how to improve the work.
>
> Evaluating the method on another, public dataset would indeed be beneficial.
> The difficulty lies in finding or creating a 4D MRI dataset with sufficient temporal (and spatial) resolution.
> Due to the nature of MRI, existing 4D MRI datasets typically use retrospective sorting, potentially in combination with heavy undersampling, to achieve a high enough temporal resolution to visualize different phases of a respiratory cycle.
> However, these datasets fail to capture the variability between respiratory cycles and, more crucially, simulating our method on them requires a higher temporal resolution still, so heavy interpolation between the already low-quality images is necessary, lowering the quality and realism of the data further.
> Therefore, we have decided not to include such a benchmark dataset.
> However, we agree that it would be a valuable bonus to apply our method to a suitable benchmark dataset in the future.
> Thank you again for your time and feedback.

---

### Official Review · Reviewer_ttZJ · 2024-02-29

**Confidence:** 4
**Preliminary Rating:** 2
**Final Rating:** 4

**Summary:**

This paper develops an RNN that predicts the rigid 3D motion of a spherical tumor from successive spokes of radial MR acquisitions to be used in MR-guided radiotherapy.  The network is trained using data simulated from an MR phantom, and experiments are performed on data simulated using both a kooshball and stack-of-stars trajectory. Experiments on simulated data show that the proposed method yields lower motion estimate errors than a constant motion estimate and achieves results slightly worse than a registration-based estimator with a prohibitive inference time. Experiments further show that the RNN trained on data simulated with the kooshball trajectory outperforms the RNN trained on data simulated with the stack of stars trajectory.

**Strengths:**

- The proposed method is simple, easy to implement, and clearly explained
- The proposed simulation procedure cleverly/carefully avoids interpolation errors in the generated data by appropriately rotating the image space phantom such that the k-space spoke at each time step is effectively “on-grid”

**Weaknesses:**

- All experiments are on simulated data (perfectly spherical tumors that are only moved by respiratory motion, without rotations) and it is unclear how well the results would translate to more realistic data
- The method only outperforms one extremely simple baseline (constant motion prediction) and does not quite match the more sophisticated registration-based baseline
- The paper is missing ablation studies for several components of the method
- The paper is missing more sophisticated analysis of the results that would help the reader understand when it fails and succeeds

**Detailed Comments:**

- “The pixel intensities of different organs and tissues for this step were extracted from T1-weighted MR images from the Duke Liver Dataset” — does this mean that every pixel within a given organ was assigned the same intensity value? If so, this is an extremely artificial simulation setup — to be more realistic, pixel intensities should be assigned probabilistically across tissues according to observed intensity distributions.
- “A spherical tumor with a random diameter between 10 and 20 mm was inserted at a different location inside the liver.” — to me it seems unlikely that this setup captures the complexity of the task that must be performed in a realistic setting. In particular, this setup along with the respiratory-only nature of the motion reduces the measured motions to translations in two dimensions. In practice, for a tumor with non-spherical shape, it would be important to estimate rotations, as this would affect how different parts of the tumor move through space, and estimating rotations significantly increases the complexity of the motion prediction task.
- Section 2.3: “Each fold consists of the 10 full sequences of two different body models, resulting in 50 sequences per fold.” Should this be “20 sequences per fold”?
- As far as I can tell, there are two unusual/unproven components to this method over a naive RNN implementation: the averaging of the symmetric spoke halves and the subsequence learning strategy. Ablation studies are needed to confirm that these aspects are truly improving the performance.
- Table 1 provides only a coarse representation of the quality of the results. There is substantial overlap between the confidence intervals for different methods, making it difficult to tell whether the observed differences are meaningful, as these large confidence intervals could be due to sequence-to-sequence variation in motion estimation quality or due to legitimate variance in the quality of the estimated motion parameters relative to the other methods. Some sample suggestions for better analyzing the results include: (1) Plotting true vs observed motion parameters as a scatterplot, with different methods in different colors, to understand whether there is a trend in whether small or large motions are miscalculated, and how consistently the proposed method outperforms the baseline. (2) A plot where the x-axis is the index of the sequence in a test subset, and the y-axis shows a box plot of distribution of error values for that sequence. This will help understand whether the error comes from a few sequences where the RNN gets “off” the prediction and then never recovers or whether errors are evenly distributed across sequences. (3) Plotting the difference between perfect and the RNN method as a function of each of the simulation parameters, to determine which cause the observed failures.

**Justification Of Final Rating:**

The paper provides an intuitive, well-explained method, and the authors have made substantive edits to the paper including extensions to 3D motion, additional ablation studies that confirm the importance of a few components of the method, and improved visualization which makes it easier to see patterns in the error of the method. I thus recommend to accept the paper, but will keep my review at "Weak Accept" due to the highly simulated nature of the data that is used to test the method (perfectly spherical tumor, artificially constant intensity across organs except for uniform noise).

**Justification Of The Preliminary Rating:**

While the proposed method is simple and easy-to-implement, its benefits are only demonstrated in a simulation setup that abstracts away significant difficulties in the realistic version of the problem. Further, it is hard to realistically evaluate how well the method is doing, as it only shows improvements compared to a weak baseline with large variance in the results. It is possible that this variance is due to, e.g. inter-subject variability, but if so, much more careful analysis is needed to show a robust improvement, especially in such a simulated setting.

**Questions To Address In The Rebuttal:**

- Was every pixel within a given organ was assigned the same intensity value?
- How do we know this method would extend to a more realistic setting with a non-spherical tumor where rotations matter?
- How much of a difference do the averaging of the symmetric spoke halves and the subsequence learning strategy make?
- In what circumstances does the observed method fail relative to the registration-based method?

---

> ### Author Response · Authors · 2024-03-17
>
> Dear reviewer,
>
> thank you very much for your thorough review and detailed feedback!
>
> The pixel intensities for a given organ vary between sequences based on the observed intensity distributions.
> However, each pixel within an organ was assigned the same intensity value.
> This is a simplification introduced by the XCAT phantom and one of the reasons we add noise during the data generation.
> The noise is added on the k-space data but this also results in noise in the image domain.
> This noise is global and not sampled from tissue-specific distributions, but it still adds variety within an organ.
>
> The spherical shape and therefore missing rotations are an unfortunate limitation of the XCAT phantom.
> Typical rotations of tumors in the liver are small, but important to estimate, especially for large, irregularly shaped tumors.
> In our previous work, a separate network was successfully used to estimate the rotation and since rotations here would mainly be induced by the periodic respiration, we are confident that the RNN could learn this relationship in 3D as well.
> However, we are aware of this limitation and are considering ways to simulate more realistic data in the future.
>
> The number of sequences per fold in section 2.3 was indeed a mistake and should have been 20.
>
> Ablation studies for the spoke halves and subsequences are a good idea.
> Hence, we have performed these experiments and added the results to the appendix.
> Using half spokes as input leads to similar results for a TR of 40 but improves the estimations for a TR of 4, compared to using the full spokes.
> The subsequence learning has a much larger impact.
> Without it, the RNN does not learn much at all for a TR of 4.
> Again, the additional variation with a TR of 40 leads to less of an impact of this, but it is still significant.
>
> We agree, that the analysis of the results was not as thorough as it should have been.
> We have added additional plots and analysis to the appendix to better understand the observed errors.
> This includes the suggested box plots to visualize the sequence-to-sequence variation, as well as a plot of the error over the respiratory phase.
> The box plots show that the MAE varies greatly between sequences, but that the RNN and the hypothetical perfect estimator perform similar on the same sequence for a TR of 4.
> However, the respiratory phase affects both methods very differently and is another source of variation in the MAE.
> While the RNN makes the largest errors around the end full inhale phase when the motion amplitude is the largest, the perfect estimator makes the largest errors between full inhale and full exhale phases, where the speed of motion is fastest.
>
> We have also clarified the comparison to the perfect estimator in the text a bit to highlight what its results mean and what the advantage of the low latency of our method is.
>
> Once again, thank you for your helpful comments. Our manuscript has greatly benefited from them.

---

> ### Comment · Reviewer_ttZJ · 2024-03-27
> **Thanks, I have updated my score.**
>
> Thanks for the detailed and thorough response to my and other reviewers' comments. In particular, I appreciate the effort to motion to 3D, the additional ablation studies, and the much-improved visualization.
>
> One comment about the additional results -- in Figure 4, it looks like a large number of examples are marked as outliers for sequences 2 and 7. Please add a comment in the text explaining why this may be the case.
>
> Overall, my major concerns have been addressed, although the simulation environment is still somewhat artificial (constant pixel intensity except for uniform noise within each organ, and perfectly spherical tumors). I have thus improved my score to Weak Accept.

---

### Author Response · Authors · 2024-03-22

Dear reviewers,

thank you very much for your detailed reviews and helpful feedback.
In response to the simplicity of only estimating motion in two axes, we have overwritten the XCAT tumor motion trajectories to include periodic motion in the x-axis based on the respiratory phase as well, generated new datasets with this motion and retrained the networks.
The paper and its results have been updated to reflect this change.

While the general sentiment of the paper remains the same, the additional motion axis not only solidifies the 3D aspect of our method but also improves its accuracy.
Small changes to the mean amplitude of motion caused by the change lead to slightly smaller errors in the y-direction and slightly larger errors in the z-direction for the two benchmark methods (from an MAE of 1.07 to 0.98 in y, and from 1.80 to 1.96 in z for the perfect estimator at TR 4).
Our proposed method has a similar change in the z-direction (going from an MAE of 1.74 to 1.86), but sees a large improvement for the error in the y-direction, which was previously larger than the error in the z-direction despite a smaller average motion amplitude.
Now, the MAE in y is 0.95 instead of 1.80, which also better matches the scale of the motion amplitude.
As a result, the total error is now comparable to the error caused by just the latency of the perfect estimator, which does not yet take the actual tracking error into account that real methods have.
This highlights the advantage of our approach even in terms of accuracy, since existing methods have latencies in the range of the simulated perfect estimator and will do those additional latency-induced errors.

Besides this, we have addressed your other comments in our individual responses and updated the paper accordingly.

---

### Meta-Review · Area_Chair_6z9S · 2024-04-04

**Recommendation:** Reject
**Confidence:** 5

**Metareview:**

This is an interesting work for rigid motion estimation for a spherical tumor from successive spokes of radial MR acquisitions to be used in MR-guided radiotherapy. The network is trained using data simulated from an XCAT phantom, and experiments are performed on data simulated using both a kooshball and stack-of-stars trajectory. Unfortunately I do not see much value of the results of this work rather than the developed RNN model since the MRI simulations are so far from being realistic. There are no coils involved (very important to estimate motion from a specific area from center of kspace) and the simulated data is not realistic for MRI for the purpose of motion correction. If the method is evaluated on any real MRI data including multi-channel kspace data and shown to work in such realistic scenarios, I believe it will add value and can be ready for publication.

---

### Decision · Program_Chairs · 2024-04-05

Accept (Poster)